Subject Areas:
genetics

Keywords:
cardiovascular diseases, gene polymorphisms, multiloci studies, myocardial infarction, stroke, Native Americans

Author for correspondence:
Rocío Gómez
e-mail: mrgomez@cinvestav.mx

# *ALOX5*, *LPA*, *MMP9* and *TPO* gene polymorphisms increase atherothrombosis susceptibility in middle-aged Mexicans

Rafael Camacho-Mejorado[1], Rocío Gómez[1],
Luisa E. Torres-Sánchez[2], Esther Alhelí Hernández-Tobías[3],
Gino Noris[4], Carla Santana[4], Jonathan J. Magaña[5],
Lorena Orozco[6], Aurora de la Peña-Díaz[7,8],
María de la Luz Arenas-Sordo[4],
Marco Antonio Meraz-Ríos[9] and Abraham Majluf-Cruz[10]

[1]Departamento de Toxicología, Cinvestav-IPN, Mexico City 07360, Mexico
[2]Centro de Investigación en Salud Poblacional, Instituto Nacional de Salud Pública, Cuernavaca, Morelos, Mexico
[3]Universidad Autonóma de Nuevo León, Facultad de Salud Pública y Nutrición, Monterrey, Nuevo León, Mexico
[4]Laboratorio Biología Molecular Diagnóstica, Querétaro, Qro, Mexico
[5]Departmento de Genética, INR, Mexico City, Mexico
[6]Laboratorio de Inmunogenómica y Enfermedades Metabólicas, INMEGEN, Mexico City, Mexico
[7]Facultad de Medicina, Departamento de Farmacología, Universidad Nacional Autónoma de México, Mexico
[8]Departamento de Biología Molecular, Instituto Nacional de Cardiología, Mexico City, Mexico
[9]Departamento de Biomedicina Molecular, Cinvestav-IPN, Mexico City, Mexico
[10]Unidad de Investigación Médica en Trombosis, Hemostasia y Aterogénesis, IMSS, Mexico City, Mexico

RG, 0000-0002-9653-7501; LET-S, 0000-0002-2902-9881; EAH-T, 0000-0003-1052-9648; AdlP-D, 0000-0001-8603-1497; MAM-R, 0000-0001-6748-8117

Atherothrombosis is the cornerstone of cardiovascular diseases and the primary cause of death worldwide. Genetic contribution to disturbances in lipid metabolism, coagulation, inflammation and oxidative stress increase the susceptibility to its development and progression. Given its multifactorial nature, the multiloci studies have been proposed as potential predictors of susceptibility. A cross-sectional study was conducted to

explore the contribution of nine genes involved in oxidative stress, inflammatory and thrombotic processes in 204 subjects with atherothrombosis matched by age and gender with a healthy group ($n = 204$). To evaluate the possibility of spurious associations owing to the Mexican population genetic heterogeneity as well as its ancestral origins, 300 unrelated mestizo individuals and 329 Native Americans were also included. *ALOX5*, *LPA*, *MMP9* and *TPO* gene polymorphisms, as well as their multiallelic combinations, were twice to four times more frequent in those individuals with clinical manifestations of atherothrombosis than in the healthy group. Once adjusting for population stratification was done, these differences remained. Our results add further evidence on the contribution of *ALOX5*, *LPA*, *MMP9* and *TPO* polymorphisms to atherothrombosis development in the middle-aged group, emphasizing the multiethnic studies in search of gene risk polymorphisms.

# 1. Introduction

Cardiovascular diseases (CVDs) are the leading cause of global death; low- and middle-income countries are where more than three-quarters of the deceases occur [1,2]. Myocardial infarction and stroke contribute to 85% of these deaths, with a consistent trend in middle-aged adults (18–50 years) [1,3]. Atherothrombosis, a clinical manifestation of atherosclerosis related to arterial occlusive thrombosis, is the essence of these two medical conditions [1,4]. Lifestyle behaviours such as tobacco and alcohol dependence, diet and reduced physical activity have been associated with an increased atherothrombosis risk [1,4]. Family history of metabolic and cardiovascular disorders suggests an inherited vulnerability and points out the potential contribution of genetic association studies as susceptibility predictors in complex diseases [5,6].

Atherothrombosis is a multifactorial disease that involves alterations in several biochemistry pathways: inflammatory response, coagulation, oxidative stress and lipid metabolism, among others [1]. Gene polymorphisms implicated in the inflammatory process, such as arachidonate 5-lipoxygenase (*ALOX5*) and those encoding to proinflammatory cytokines (i.e. interleukins (IL) 1, 6, 12, 15, 18, etc.), have been associated with the atherosclerotic plaque development [7,8]. *ALOX5* encodes to a key enzyme (5-lipoxygenase) in the biosynthesis of leukotrienes. The shorter repeat alleles, $(GGGCGG)_{3-4}$, of a promoter region polymorphism in *ALOX5*, have been associated with an increase in leukotriene expression and the carotid intima-media thickness [7,9]. Expression and activity of matrix metalloproteinases (MMPs) are induced in reaction to inflammatory response. The largest alleles, $(CA)_{>21}$, of MMP9 promoter polymorphism, have been associated with expression alterations, mainly, in advanced atherosclerotic lesions [10,11]. Regarding reactive oxygen species (ROS), nitric oxide (NO), produced by endothelial nitric oxide synthase (eNOS), is a crucial regulator of vascular homeostasis and a potent antiatherogenic and antithrombotic [12,13]. The *eNOS* dinucleotide polymorphism, $(CA)_n$, has been positively associated with coronary artery disease risk; the number of repeats seems to modulate the splicing efficiency, controlling the gene expression and enzymatic activity [14]. In response to oxidative stress, antioxidant enzymes, such as heme oxygenase 1 (HMOX1), increase their expression [15]. The large repeat alleles, $(GT)_{19-39}$, of the promoter region of *HMOX1*, have been linked to low enzyme activity and an increase in coronary artery disease and myocardial infarction risks [15–17]. Candidate genes related to endothelial dysfunction, such as nicotinamide adenine dinucleotide phosphate oxidases (NOX), and particularly, NOX4, have been proposed, although this matter has been hardly studied [18,19].

Once atherosclerotic plaque has been formed, platelet aggregation and the prothrombotic responses are triggered [20]. Polymorphisms in genes encoding to thyroid hormones and the von Willebrand factor have been related to platelet aggregation, thrombus formation and hypofibrinolysis, and, in turn, the risk of ischaemic stroke and coronary heart disease [21]. Previously, our research group reported the contributions of two tetranucleotide polymorphisms with venous thromboembolism: (i) in thyroid peroxidase gene (*TPO*), and (ii) in the von Willebrand factor (*vWA*) gene, suggesting a possible contribution to atherothrombosis [22].

Lipoprotein(a), Lp(a), is low-density lipoprotein attached to apolipoprotein B-100, which is linked, in turn, to apolipoprotein(a) [23]. Pentanucleotide $(TTTTA)_n$ allele variation in the controlling region of the Lp(a) gene (*LPA*) is related to variable isoforms of the kringle IV at apolipoprotein(a) and, consequently, with atherosclerosis and thrombosis risk [24,25]. Lipoprotein(a) plasma level concentrations have shown a high heritability [26].

Phenotype–genotype associations in candidate genes have been performed in different populations. Nonetheless, most (96%) have been conducted in European ancestry populations [27]. The findings

obtained in Native and mixed ancestry populations have revealed undescribed risk polymorphisms besides interethnic variability in genes associated with clinical traits; these studies represent less than 4% of phenotype–genotype associations [28,29].

The identification of early-onset CVDs susceptibility polymorphisms in the Mexican population is crucial because the morbidity and mortality rates have risen substantially in the last years [30,31]. Herein, the collective contribution of nine genes (*ALOX5, IL6, LPA, MMP9, vWA, eNOS, HMOX1, NOX4* and *TPO*) was investigated in middle-aged Mexican subjects with and without clinical manifestations of atherothrombosis.

# 2. Material and methods

## 2.1. Characteristics of the studied groups

A cross-sectional study was conducted from October 2015 to December 2016. There were 408 participants (less than or equal to 48 years old) who were unrelated men and women residents of Mexico City and selected from the General Regional Hospital No. 1 at the Mexican Institute of Social Security. The group with clinical manifestations of atherothrombosis (CMA) was identified in the Thrombosis, Haemostasis and Atherogenesis Medical Research Unit, including 204 subjects with a confirmed diagnosis of myocardial infarction (42%) and ischaemic stroke (58%). Each subject with CMA was matched by age (±5 years) and gender with one healthy individual selected from the blood bank (*n* = 204) from the same hospital. Classical risk factors such as diabetes type 2, hypercholesterolaemia, body mass index, age, use of hormones (in women) and family history of CVDs were obtained from questionnaire before the sampling in these 408 individuals (electronic supplementary material, table S1).

Dietary, physical activity, smoking habit and alcohol intake were obtained through a direct interview. Diet was assessed with 24 h questionnaire using the previous day's eating habits. Based on the food type and the eaten portion reported, calorie and macronutrients (i.e. carbohydrates, lipids and proteins) intake were estimated using the Mexican System of Food Equivalents [32]. Energetic expenditure associated with the daily physical activities performed at work, recreational activities, fitness, sports, etc. were estimated using the World Health Organization questionnaire (www.who.int/ncds/surveillance/steps/GPAQ_ES.pdf) [33].

Smokers and ex-smokers cumulative cigarette consumption were expressed by pack-years until the occurrence of disease. The smoking index was obtained by multiplying the number of cigarettes smoked per day by the smoking duration; the obtained value was divided by 20. Depending on the smoking index mean, the subjects were categorized as follows: (i) individuals consuming less than two cigarette packs per year and (ii) individuals consuming more than two cigarette packs per year. Non-smokers were considered as the reference category.

Alcohol consumption was expressed in grams (g week$^{-1}$). Based on the alcohol consumption mean obtained (20 g week$^{-1}$), people who reported alcohol consumption was categorized such as those who drink less than 20 g week$^{-1}$ and who more than 20 g week$^{-1}$; abstemious individuals (0 g week$^{-1}$) were the reference group.

Two additional groups were included to control the bias caused by the recent rise of the Mexican mestizo population (approx. 500 years ago): genomic control (GC) and ancestral control (AC). GC was included because the Mexican mestizo population (the most prominent Mexican ethnic group) presents a complex genetic architecture both by its youth and by the heterogeneity of the parental populations from which it arose (Native Americans, Europeans and African slaves). These two conditions (youth and genetic heterogeneity) could be generators of phenotype–genotype spurious associations [34–36]. GC was a random sample of 300 unrelated individuals (150 men and 150 women) that belong to the Mexican mestizo population with at least three generations of ancestors born in this country, who share an ethnic origin with the subjects with CMA and the healthy group. GC samples were obtained from the Central Valley of Mexico (including the states of Guanajuato, Mexico City, Puebla and Queretaro) places of birth reported by CMA and healthy subjects. Of note, each one of the GC subjects has a genetic profile analysis with 21 neutral polymorphisms belonging to the combined DNA index system, guaranteeing that there was no genetic relationship among the individuals included [37,38]. The differences between GC and the healthy group are that in the latter, the atherothrombotic status is known and exhibits homogeneity about their CVDs physiological condition (disease absence). In the GC group, the atherothrombotic status is unknown and possibly contains several subpopulations with different allele frequencies and disease prevalence. The allele

frequency distribution in GC using the nine polymorphisms allows for the estimation of the variance inflation factor caused by the population stratification and corrects it.

AC was integrated by 329 Native Americans belonging to seven ethnic groups (Maya, Mazahua, Me'Phaas, Mixe, Nahua, Rarámuris and Yoremes) from four different linguistic families (electronic supplementary material, table S2). Native American ancestry is one of the most prominent in the central region of Mexico [39]. Determination of the allele frequency distribution in the GC and AC groups was used to avoid the influence of the ancestral background on the frequency differences found between the groups with CMA and the healthy.

Ethics Committees of the Mexican Institute of Social Security (IMSS, initials in Spanish; approved number R-2015-3609-20), the Centre of Research and Advanced Studies from the National Polytechnic Institute, the National Institute of Genomic Medicine and the Laboratory of Molecular Biology Diagnostic approved the protocol and sample collection. All participants signed an informed consent form validated by the Ethics Committees. The present study was conducted in agreement with the principles established by the Declaration of Helsinki.

## 2.2. DNA isolation

The 408 samples collected for the present study were obtained through EDTA anticoagulant blood collection tubes using a BD Vacutainer system (Becton Dickinson, Franklin Lakes, NJ, USA). From these, genomic DNA was isolated from the buffy coat with Blood DNA Preparation Kit (Jena Bioscience, Jena, Thuringia, Germany). The purity and DNA concentrations were evaluated using NANODROP 2000 (Thermo Fischer Scientific, Suwanee, GA, USA); agarose (Amresco, Solon, OH, USA) gels 0.8% were used to evaluate DNA integrity. GC and AC samples were obtained from prior studies of our research group.

## 2.3. Polymorphisms explored

A total of nine short tandem repeat polymorphisms were explored in the 1037 individuals studied. Of these, *ALOX5*, *IL6*, *LPA* and *MMP9* gene polymorphisms were involved in inflammatory processes [9,24,40–43]. Another three gene polymorphisms (*eNOS*, *HMOX1* and *NOX4*) were involved in oxidative stress processes, and the *vWA* and *TPO* gene polymorphisms were involved in the coagulation mechanisms [13–16,22,44]. The detailed information about these nine polymorphisms is depicted in electronic supplementary material, table S3.

## 2.4. Determination of genotypes

The nine polymorphisms were amplified with an endpoint polymerase chain reaction using Kyratec Supercycler SC200 thermal cycler (Queensland, Australia). Three multiplex amplifications and one by separate were performed for simultaneous genotyping. The four susceptibility polymorphisms (*ALOX5, LPA, MMP9* and *TPO*) were further included in a simultaneous amplification (electronic supplementary material, table S4).

To avoid the false positive rates associated with the ABO system, blood phenotypes were also performed from endpoint multiplex reactions as in Jiang *et al.* [45]. All amplicons were obtained with a thermal cycler Kyratec Supercycler SC200 thermal cycler (Queensland, Australia).

The genotype assignations of the nine polymorphisms were obtained by capillary electrophoresis analysis (ABI Prism 3130XL Genetic Analyser) using the validated software GeneMapper ID v. 3.2 (Applied Biosystems, Carlsbad, CA, USA).

## 2.5. Statistical analyses

### 2.5.1. Differences between the CMA and the healthy group

Subjects with and without CMA were compared according to selected characteristics. Depending on the variable, Student's *t*-tests, $\chi^2$ tests or Fisher exact tests were used.

Genetic associations with the CMA were estimated using unconditional logistic regression models. Allelic and genotype associations were assessed using the following risk categories: *ALOX5* genotype carriers 3/(3), with (3) being any different allele to risk allele; *LPA* carriers genotype (8)/(8), with (8) being any different allele to the allele 8; *MMP9* genotype carriers ≤14/≤14, and *TPO* genotype carriers 8/8.

Age at the interview was included as a continuous variable in bivariate and multivariable models. The final models were adjusted as follows: the personal history of hypertension and/or diabetes, family records of CVDs, smoking history, alcohol intake and physical activity. All analyses were performed using STATA v. 14 statistical software (StataCorp LP, College Station, TX, USA).

### 2.5.2. Genetics statistics

Allele and genotype frequencies, the number of different alleles ($k$), observed (Ho) and expected heterozygosity (He) and linkage disequilibrium (LD) were estimated with Arlequin v. 3.5 [46]. Hardy–Weinberg (HW) expectations were obtained using Weir and Cockerham's $F$ statistics with Genètix v. 4.05.2 [47]. Multiallelic combination frequencies were determined by direct counting considering the cumulative effect concerning the risk alleles of each studied polymorphism. The genetic relations among the studied groups were assessed with the genetic distances ($R_{ST}$) using 1000 permutations, followed by analysis of molecular variance with Arlequin v. 3.5 [46] and visualized by the multidimensional scaling (MDS) plot using SPSS v. 23 (IBM Corp., Armonk, NY, USA).

### 2.5.3. Estimation of population subgroups

The presence of population subgroups was inferred from the genotypes obtained with the nine polymorphisms studied. As mentioned before, the population's genetic heterogeneity might cause spurious associations linked with the genetic background rather than related to the disease. The presence of distinct subgroups within the groups studied was inferred with Bayesian methods using Structure v. 2.3.3 [48] following the conditions described by Gomez et al. [49]. The results obtained were analysed with Structure Harvester using the method described by Evanno [50] (http://taylor0. biology.ucla.edu/structureHarvester/). This method infers the number of population subgroups, which then was used to adjust the difference of frequencies between the individuals with CMA and the healthy group using Strat v. 1.1 [51]. Those polymorphisms that maintained the statistical differences after the stratification correction were involved in further analyses.

# 3. Results

## 3.1. Characteristics of the studied groups

Because of the study design, no differences between age and gender features were found. Nonetheless, classical risk factors such as hypertension (27.9 versus 17.6%; $p = 0.01$), hypercholesterolaemia (20.1 versus 6.4%; $p < 0.01$) and family history of CVDs (54.9 versus 27.4%; $p \leq 0.0001$) were more frequent in CMA than in the healthy group. Tobacco index and alcohol intake also presented differences. Regarding the smoking index, the CMA group had the biggest share of individuals smoking or having smoked more than two cigarette packs per year ($p = 0.004$). By contrast, alcohol consumption habits were higher in the healthy group ($p = 0.02$; table 1).

## 3.2. Distribution of allele and genotypes in CMA and healthy groups

The frequency differences between CMA and the healthy group for the risk alleles and its contribution to atherothrombosis are shown in table 2. The frequency differences for all alleles between CMA and the healthy group are shown in electronic supplementary material, tables S5 and S6. The allele 3 in ALOX5 (ALOX5*3) and the shorter alleles, (CA)≤14, in MMP9 were associated with CMA risk; analogous results were observed with the genotype frequencies ($p < 0.001$). The non-carriers genotypes of the 8-allele (8)/(8) in LPA ($p = 0.004$), and the homozygous genotype 8/8 in TPO ($p < 0.001$) were more frequent in CMA than in the healthy group, exhibiting a genetic contribution to atherothrombosis (tables 2 and 3).

Regarding the IL6, vWA, eNOs, HMOX1 and NOX4 polymorphisms, no differences in the allele frequencies were found. Nevertheless, flimsy trends were presented, such as IL6 (5/5 and 5/7), eNOS (28/30, 29/32 and 32/32), HMOX1 (21/30) and NOX4 (15/18). Of note, the homozygous genotype 17/17 in vWA showed a possible contribution (OR = 2.23, 95% CI = 1.01–4.34, $p = 0.03$). The differences found in these five polymorphisms were lost after the comparison with GC.

The genotype frequencies in all gene polymorphisms studied are shown in electronic supplementary material, tables S7–S15.

**Table 1.** Characteristics of the studied participants. s.d., standard deviation; METs, metabolic equivalents; CVDs, cardiovascular diseases. Italicized numbers depict significant differences between the studied groups.

| | clinical manifestations of atherothrombosis[a] | | |
| | yes | no | |
| characteristics | $n = 204$ (%) | $n = 204$ (%) | $p$-value[b] |
|---|---|---|---|
| age | | | |
| mean ± s.d. | 47.5 ± 14.5 | 48.0 ± 16.6 | ≥0.05 |
| gender (%) | | | |
| male | 102 (50) | 102 (50) | 1 |
| female | 102 (50) | 102 (50) | |
| personal pathological history | | | |
| diabetes (%) | | | |
| yes versus no | 27 (13.20) | 21 (10.30) | ≥0.05 |
| hypertension (%) | | | |
| yes versus no | 57 (27.90) | 36 (17.60) | 0.01 |
| hypercholesterolaemia | | | |
| yes versus no | 41 (20.10) | 13 (6.40) | *≤0.01* |
| smoking index (pack-year) | | | |
| never smoker | 94 (46.10) | 121 (59.30) | |
| ≤2 | 40 (19.60) | 42 (20.60) | *>0.01* |
| >2 | 70 (34.30) | 41 (20.10) | |
| physical activity (METs week$^{-1}$) | | | |
| none | 54 (26.50) | 43 (21.10) | |
| ≤1480 | 73 (35.80) | 81 (39.70) | ≥0.05 |
| >1480 | 77 (37.70) | 80 (39.20) | |
| body mass index (%) | | | |
| normal | 71 (35.10) | 68 (33.30) | |
| overweight | 95 (47) | 91 (44.60) | ≥0.05 |
| obesity | 36 (17.80) | 45 (22.10) | |
| alcohol intake (g week$^{-1}$) | | | |
| never | 92 (45.10) | 80 (39.20) | |
| ≤20 | 38 (18.60) | 64 (31.40) | *0.02* |
| >20 | 74 (36.30) | 60 (29.40) | |
| family history of CVDs, positive (%) | | | |
| yes versus no | 112 (54.90) | 56 (27.40) | *≤0.01* |

[a]Myocardial infarction and stroke.

[b]$\chi^2$ and Student's $t$-tests.

## 3.3. Comparison between CMA and the healthy groups with the GC

An irrelevant genetic variation (4%; non-significant $p$-value), between the CMA and healthy groups with the GC, was found, confirming that these three groups share genetic characteristics (figure 1; electronic supplementary material, table S16). This finding was consistent with the genetic distances ($R_{ST}$) where no differences between the healthy group and GC were found (non-significant $p$-value). By contrast, the genetic distance in the CMA group was significantly different from the healthy and GC groups ($p = 0.002$, $p = 0.030$, respectively). The differences between CMA and GC were mainly related to

**Table 2.** Contribution of the genetic factors and clinical manifestations of atherothrombosis. OR, odds ratio; CI, confidence interval. Italicized numbers depict significant differences between the studied groups. The number between the parentheses means a different allele to the risk one for ALOX5, MMP9 and TPO loci. In LPA, the genotype (8)/(8) represents non-carriers of the allele 8, with (8)/(8) being the risk condition.

| locus | genetic variant | | clinical manifestations of atherotrombosis | | clinical manifestations of atherothrombosis versus healthy controls | | | | | |
| | | | yes[a] n = 204 (%) | no n = 204 (%) | OR (95% CI)[b] | p-value | OR (95% CI)[c] | p-value | OR (95% CI)[d] | p-value |
|---|---|---|---|---|---|---|---|---|---|---|
| **ALOX5** | allele 3 | yes | 20 (4.90) | 4 (1) | *5.21 (1.76–15.40)* | *0.03* | *5.51 (1.82–16.70)* | *0.03* | *5.03 (1.63–15.50)* | *<0.01* |
| | | no | 388 (95.10) | 404 (99) | 1 | | 1 | | 1 | |
| | genotype | 3/3 + 3/(3) | 16 (7.80) | 4 (2) | *4.26 (1.94–9.37)* | *<0.01* | *4.21 (1.87–9.46)* | *<0.01* | *3.76 (1.63–8.60)* | *<0.01* |
| | | (3)/(3) | 188 (92.20) | 200 (98) | 1 | | 1 | | 1 | |
| **LPA** | allele 8 | no | 102 (25) | 81 (19.80) | 1.35 (0.97–1.87) | 0.08 | 1.33 (0.94–1.87) | 0.11 | 1.33 (0.93–1.91) | 0.12 |
| | | yes | 306 (75) | 327 (80.20) | 1 | | 1 | | 1 | |
| | genotype | (8)/(8) | 91 (44.60) | 71 (34.80) | *1.51 (1.14–2.01)* | *<0.01* | *1.47 (1.10–1.98)* | *0.01* | *1.50 (1.11–2.04)* | *0.01* |
| | | 8/8 + (8)/(8) | 113 (55.40) | 133 (65.20) | 1 | | 1 | | 1 | |
| **MMP9** | alleles ≤ 14 | yes | 121 (29.70) | 82 (20.10) | *1.69 (1.22–2.34)* | *<0.01* | *1.52 (1.09–2.13)* | *0.01* | 1.42 (1.00–2.01) | 0.05 |
| | | no | 287 (70.30) | 326 (79.90) | 1 | | 1 | | 1 | |
| | genotype | ≤14/≤14 | 17 (8.30) | 5 (2.40) | *3.66 (1.78–7.52)* | *<0.01* | *2.86 (1.36–6.02)* | *<0.01* | *2.54 (1.18–5.48)* | *0.02* |
| | | (≤14)/≤14+(≤14/≤14) | 187 (91.70) | 199 (97.60) | 1 | | 1 | | 1 | |
| **TPO** | allele 8 | yes | 219 (53.70) | 199 (48.80) | 1.22 (0.92–1.60) | 0.16 | 1.13 (0.84–1.50) | 0.41 | 1.11 (0.83–1.50) | 0.48 |
| | | no | 189 (46.30) | 209 (51.20) | 1 | | 1 | | 1 | |
| | genotype | 8/8 | 79 (38.70) | 49 (24) | *2 (1.48–2.71)* | *<0.01* | *1.81 (1.32–2.48)* | *<0.01* | *1.66 (1.20–2.30)* | *<0.01* |
| | | 8/(8) + (8)/(8) | 125 (61.3) | 155 (76) | 1 | | 1 | | 1 | |

[a]Myocardial infarction and stroke.
[b]Adjusted by age and sex.
[c]Adjusted by age, sex and family history of CVDs.
[d]Adjusted by age, sex and family history of CVDs, smoking history, alcohol intake and physical activity.

**8**

**Table 3.** Comparison of the distribution of the genetic factors between the subjects with clinical manifestations of atherothrombosis and GC. CMA, clinical manifestations of atherothrombosis; OR, odds ratio; CI, confidence interval. The number between the parentheses means a different allele to the risk one for *ALOX5*, *MMP9* and *TPO loci*. In *LPA*, the genotype (8)/(8) represents non-carriers of the allele 8, with (8)/(8) being the risk condition. Italicized numbers show the statistical significance after population stratification correction.

| locus | genetic variant | | CMA[a] n = 204 (%) | genomic control n = 300 (%) | OR (95% CI) | p-value |
|---|---|---|---|---|---|---|
| **ALOX5** | allele 3 | yes | 20 (4.90) | 13 (2.16) | 2.33 (1.14–4.73) | *<0.01* |
| | | no | 388 (95.10) | 587 (97.83) | 1 | |
| | genotype | 3/3 + 3/(3) | 16 (7.80) | 12 (4) | 2.04 (1.18–3.52) | *<0.01* |
| | | (3)/(3) | 188 (92.20) | 288 (96) | 1 | |
| **LPA** | allele 8 | no | 102 (25) | 112 (18.67) | 1.45 (1.07–1.97) | *>0.05* |
| | | yes | 306 (75) | 488 (81.33) | 1 | |
| | genotype | (8)/(8) | 91 (44.60) | 103 (34.33) | 1.54 (1.19–1.99) | *>0.05* |
| | | 8/8 + (8)/(8) | 113 (55.40) | 197 (65.67) | 1 | |
| **MMP9** | alleles ≤14 | yes | 121 (29.70) | 135 (22.50) | 1.45 (1.10–1.93) | *>0.05* |
| | | no | 287 (70.30) | 465 (77.50) | 1 | |
| | genotype | ≤14/≤14 | 17 (8.30) | 16 (5.33) | 1.61 (0.98–2.66) | *>0.05* |
| | | (≤14)/≤14 + (≤14/≤14) | 187 (91.70) | 284 (94.67) | 1 | |
| **TPO** | allele 8 | yes | 219 (53.70) | 302 (50.33) | 1.14 (0.89–1.47) | *>0.05* |
| | | no | 189 (46.30) | 298 (49.67) | 1 | |
| | genotype | 8/8 | 79 (38.70) | 75 (25) | 1.89 (1.44–2.49) | *0.02* |
| | | 8/(8) + (8)/(8) | 125 (61.3) | 225 (75) | 1 | |

[a]Myocardial infarction and stroke.

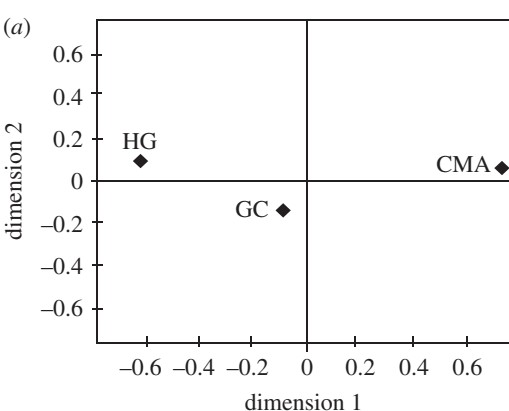 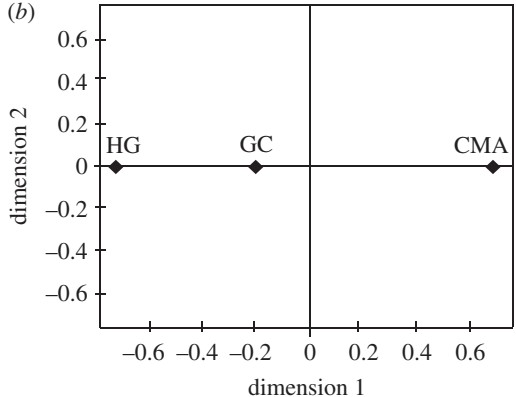

**Figure 1.** Multidimensional scale plot of genetic distances ($R_{ST}$) values estimated from (*a*) all polymorphisms explored and (*b*) only with the four susceptibility polymorphisms. CMA, clinical manifestations of atherothrombosis; GC, genomic control; HG, healthy group.

*ALOX5* (4.9 versus 2.16; $p \leq 0.0001$) and *TPO* (38.7 versus 25; $p \leq 0.0005$) (table 3). These dissimilarities persisted after population stratification readjustment ($p = 0.02$).

Multiallelic combination frequencies comparison, including *ALOX5, LPA, MMP9* and *TPO*, was twice as inferior in GC than the CMA group (data not shown). Thus, a clear separation between the CMA group and the rest of the groups was depicted in the MDS plot (figure 1).

## 3.4. Genetic statistical parameters

HW expectations and LD for all studied polymorphisms are depicted in electronic supplementary material, tables S5, S6 and S16–S23. The healthy group and the GC were in HW and linkage equilibrium, whereas the CMA group presented a remarkable HW departure ($p \leq 0.0001$) related to homozygote excess ($F_{IS} > 0$). These differences were exhibited in *ALOX5* (Ho = 0.270 versus He = 0.336; $F = 0.197$) and *TPO* (Ho = 0.422 versus He = 0.642; $F = 0.342$) gene polymorphisms. HW departure in both polymorphisms was conserved even after Bonferroni's *post hoc* test ($p \leq 0.006$).

## 3.5. Native populations genetic characterization

A notable drop in the frequency of *ALOX5* and *MMP9* risk alleles was observed in the Native peoples (electronic supplementary material, tables S16–S23). It is worth mentioning that the genotype 8/8 in *TPO* showed an interethnic variation (electronic supplementary material, table S14). Nahuas and Me'Phaas presented the highest values, whereas Mayas and Raramuris exhibited the lowest. The frequency of (8)/(8) in *LPA* was lower in Natives than the CMA group (electronic supplementary material, table S11).

## 4. Discussion

The identification of susceptibility polymorphisms in genes involved in diverse biochemical pathways provides the best risk predictor to a better understanding of the pathogenesis of the chronic heritable diseases. CVDs are polygenic disorders with an estimated heritability of 40–50%; its incidence in Mexican middle-age adults is the third cause of death [3,30,52]. In the present cross-sectional study, the polymorphisms of nine genes involved in inflammation, coagulation disorders and oxidative stress were evaluated in a group with CMA matched with a healthy group. Frequency differences in alleles, genotypes and multiallelic combinations were found in *ALOX5, LPA, MMP9* and *TPO* polymorphisms, suggesting a possible contribution to CVDs development.

*ALOX5\*3* was significantly more frequent in the subjects with CMA; prior findings have corroborated our results in African American, European and Latino populations [7,9,53–56]. Such contribution has been related to the low transcriptional activity exhibited by the shorter alleles (i.e. *ALOX-5\*3*), altering the biosynthesis of leukotrienes [9,57]. The relationship between the low number of tandem repeats (shorter alleles) and the increased risk of CMA is in accordance with the results obtained in animal

and cell models [9,53,54,57]. The pathologic evidence (post-mortem) has demonstrated a positive correlation between increased *ALOX5* expression with several stages of atherosclerotic lesions; the advanced lesions presented the most notable increase [58]. It was of particular note that Native American people presented the lowest frequencies of the risk allele (*ALOX5*3*), supporting the variation across ethnicities reported previously [9,53]. These reports show that the Asian and Afrodescendant populations exhibit the highest frequencies [9,53]. The closeness between the Native Americans and the Asians is contrary to these findings [59]. Nonetheless, the Americas' colonization involved a prominent genetic bottleneck, which may support the low frequencies of *ALOX5*3* in the inbreeding Indigenous populations [60]. The well-preserved association with *ALOX5*3* even after the adjustment of genetic stratification and classical risk factors reinforces our results. Some discrepancies have been found in different reports, including genome-wide association studies [61,62]. These disparities could be explained by the frequency differences across populations and by the genome-wide association studies' focus on common variants while *ALOX5*3* is a low-frequency allele.

*LPA* and *MMP9* have been recognized as risk factors, *per se*, to atherosclerosis development via lipid metabolism, inflammation and oxidative stress [4]. Thus, it is not surprising to find significant associations within the polymorphisms of these two genes with CMA, which have been replicated by epidemiological, cohort and meta-analysis studies [25,63,64]. As mentioned before, Lp(a) has a protein moiety apoB-100 linked by a disulfide bridge to multi-kringle structure, also called apolipoprotein(a) [23,65]. Apolipoprotein(a) and plasminogen have a robust structural similarity, competing for fibrin affinity sites (lysine residues) and promoting the free radicals generation [24,65]. The smaller, $(TTTTA)_{<7}$, and larger, $(TTTTA)_{>8}$, alleles have been associated with shorter kringle IV repeat numbers and, in turn, with high Lp(a) plasma concentrations and prothrombotic activity, reproducing our findings [24,26,65].

*MMP9* encodes a zinc-binding endopeptidase with activity to degrade extracellular matrix and collagen fibres, causing plaque destabilization, free radicals generation and proinflammatory cytokines activation [10]. *MMP9* is mainly regulated at the transcriptional level; polymorphisms located in the promoter region have been associated with myocardial infarction, ischaemic stroke, coronary artery disease, lung disorders and a variety of carcinomas (i.e. gastric, breast, urinary) [11,64,66,67]. Short alleles (*MMP9*14*) in epithelial cells have shown increased expression activity (two and threefold more) compared to than those with larger repeats (*MMP9*20*) [68]. It is noteworthy that MMP9 expression is induced by proinflammatory cytokine properties presented by *ALOX5* and *LPA*, illustrating the multifactorial nature of CVDs [4,11,26,64].

Although *LPA* and *MMP9* have presented consistent contributions to CVDs development, controversial results attributable to the ethnic and individual variation have also been documented. Different Lp(a) serum levels, possibly related to the genetic heterogeneity, have been reported in the Mexican population [69,70]. Also, ethnic variations in the *MMP9* allelic distribution have been reported; European populations exhibited the highest frequencies of the *MMP9*14* allele; *MMP9*21* is the most prominent one within Asian ancestry [63,64,68,71]. A prior study of our research group described a bimodal *MMP9* distribution in the Mexican *mestizos* (alleles 14 and 21), whereas Native Americans presented similar distributions to Asian populations [29].

The ethnic variation within the Mexican population could explain the loss in the *LPA* and *MMP9* contribution once population stratification was corrected. However, these findings do not contradict our results. By contrast, they delineate the importance of multiethnic studies and its relevance in search of gene polymorphisms contributing to complex disease development. In the light of this evidence, a finer-scale multiethnic study has reported more than 100 undescribed *LPA* gene polymorphisms, which could explain the relevance of our research and the dissimilarities across populations [26].

*TPO* encodes a key enzyme for the generation of thyroid hormones (TH). Numerous studies have associated the modification in thyroid function (i.e. hypo and hyperthyroidism) with several effects on cardiovascular system functions [72]. Deficient levels of TH cause hypercholesterolaemia and increase the carotid intima-media thickness. High TH levels cause hypercoagulation and hypofibrinolysis, increasing thrombus formation; both conditions cause hypertension. All these clinical manifestations contribute substantially to atherothrombosis development. In the light of this evidence, TH alterations have been associated with diverse CVDs [22,73–75].

In connection with the number of *TPO* tandem repeats, the genotype 8/8 exhibited a remarkable association with CMA. The studies that reinforce this contribution are scanty; however, in a prior study, we have reported a significant association between *TPO* and venous thrombosis risk [22]. Another study has shown a nuanced contribution (13%) of *TPO*-8/8 genotype with autoimmune

thyroiditis risk, which is mainly associated with hypothyroidism [76]. Of note, the allele 8 is the most frequent one in worldwide populations; nonetheless, the homozygous genotype frequencies present differences in terms of ethnic diversity [77]. In a prior report, the genotype differences between autoimmune thyroiditis individuals and its healthy controls were sparse; herein, the frequency of risk genotype was 1.6-fold higher in the CMA group than in the healthy subjects. The remarkable HW departure found in *TPO*, along with the frequency differences between the groups, fortifies our results. Accordingly, other genetic determinants within the *TPO*, as in other genes involved in the thyroid function, have shown association with cardiovascular risk, suggesting a genetic influence through TH levels [74,75].

A possible explanation to the contribution of the *TPO* intronic polymorphism to the CMA could be alternative splicing [78]. Note, four out of eight *TPO* isoforms recently related to breast cancer lacked intron 10, where *TPO* polymorphism is contained [79]. Interestingly, these isoforms have shown a partial decrease in *TPO* expression, impacting in TH production; similar conditions could be occurring in CVDs. Despite the recommendations of the American Heart Association to evaluate thyroid functions in cardiovascular patients, this practice is not periodically used in Mexico [80]. It is noteworthy that in both underlying cardiac status, as in the subclinical thyroid dysfunctions, the TH levels are affected [81–83]. The Mexican population has shown a high prevalence of thyroid dysfunction (approx. 24%), with subclinical hypothyroidism being the most common condition (15–18%), whose recurrence is up to 1.8-fold increase with respect to other reports (3–10%) [72,84,85]. Possibly, TH's normal ranges do not reflect subclinical conditions and, in turn, the inherent cardiovascular risk of the population. Hence, the thyroid function should be defined through THs levels in the general population, as in health and disease conditions, and by age, sex and ethnicity [86].

The results obtained in *IL6*, *vWA*, *eNOS*, *HMOX1* and *NOX4* polymorphisms did not show significant contributions. Nonetheless, it does not discard their possible contribution to CVDs risk, which is a polygenic condition.

The validity of our results was sustained by the synchronic exploration of the nine polymorphisms and involved in different biological processes. The relevance of considering the genetic background of the populations studied to lessen the statistical error type I was another of the strengths of the present study. HW and linkage equilibriums in all polymorphism and all groups analysed, excepting those where a stronger genetic contribution was found (*ALOX5* and *TPO*), also support the validity of our results. Finally, the possibility of genotyping errors seems unlikely, as the gene distributions are fairly close to those reported by prior studies. Nonetheless, the scantiness of genetic data to compare our findings is a possible weakness; thus, our results should be interpreted in the light of this limitation.

# 5. Conclusion

Our results add to evidence regarding the role of *ALOX5*, *LPA*, *MMP9* and *TPO* gene polymorphisms, and their contribution to atherothrombosis development. The low frequencies within the risk alleles supported the role of the rare variants in the aetiology of the complex diseases. To our knowledge, this is the first study in the Mexican population where ancestral and genomic controls reinforce the relevance of the role of these polymorphisms in atherothrombosis development.

Ethics. The Ethics Committees of the Mexican Institute of Social Security (IMSS, initials in Spanish; approved number R-2015-3609-20), the Centre of Research and Advanced Studies from the National Polytechnic Institute, the National Institute of Genomic Medicine, and the laboratory of Molecular Biology Diagnostic approved the protocol and sample collection. All participants signed an informed consent form validated by the Ethics Committees. The present study was conducted in agreement with the principles established by the Declaration of Helsinki.
Data accessibility. The datasets supporting this article have been uploaded as part of the electronic supplementary Material.
Authors' contributions. R.C.-M., R.G. and A.M.-C. designed the study and coordinated and conceived the clinical research. R.C.-M., E.A.H.-T., C.S., G.N., J.J.M. and L.O. conducted the sample collection and performed the genotyping and acquisition of data. R.G., L.E.T.-S., R.C.-M., C.S., G.N., M.A.M.-R. and M.d.l.L.A.-S. performed the statistical analyses and interpretation of data. R.G. wrote the manuscript. All authors reviewed and approved the final version of the document. All authors gave final approval for publication.
Competing interests. The authors declare no competing interests and no financial relationship with the organization that sponsored the present research.
Funding. The present study was financed by the National Council of Science and Technology (Conacyt, acronym in Spanish), through the grant nos. 261268 and 178239 both to R.G. and the scholarship 347076 to R.C.-M. during his PhD.

Acknowledgements. We thank the Genomics, Proteomics and Metabolomics Laboratory from LaNSE-Cinvestav for helping with the genotyping processes. We are grateful to all the participants for their enthusiastic collaboration as well as to the work team of the Thrombosis, Hemostasis and Atherogenesis Medical Research Unit (Carlos McGregor Hospital, IMSS) for their help in the sampling.

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
