## [Reviewer comments · Royal Society Open Science]

Review History

RSOS-190775.R0 (Original submission)

Review form: Reviewer 1

Is the manuscript scientifically sound in its present form?

Yes

Are the interpretations and conclusions justified by the results?

Yes

Is the language acceptable?

No

Is it clear how to access all supporting data?

Not Applicable

Do you have any ethical concerns with this paper?

No

Have you any concerns about statistical analyses in this paper?

I do not feel qualified to assess the statistics

Recommendation?

Accept with minor revision (please list in comments)

Comments to the Author(s)

The researchers looked for genetic associations with atherothrombosis development (clinical manifestations) in the protein-coding genomes of 408 unrelated Mexican individuals. 204 Individuals having clinical manifestations of atherothrombosis (n=204) were paired by age and gender to an equal number of blood donors without clinical manifestations. They then replicated the study in an independent group of 300 unrelated mestizo individuals, the so-called genome control, and in a group of 329 native subjects belonging to different ethnic groups.

From the nine gene studied, the researchers discovered frequency differences in 4 of them: ALOX5, LPA, MMP9, and TPO. These were significantly more frequent in individuals with clinical atherothrombosis even after being adjusted by family history of cardiovascular disease, smoking, alcohol intake, and physical activity

Main concerns

1. In the abstract the authors included both patients and control groups in a collective of "1037 unrelated Mexican individuals". The real number of patients is 204, which had clinical atherothrombosis. There is an equal number of blood donors without clinical manifestations of atherothrombosis. There are also a genome control group of 300 unrelated individuals (equality in gender) and an ancestral control group of 329 from different ethnic origin. These differences should be clearly indicated not only in the Abstract but also sufficiently detailed in section 3.1 Population studied.
2. I would like to understand which is the difference between the 204 asymptomatic blood donor controls and the so-called genome control group of 300 unrelated individuals (beside their geographical origin). I suppose that residents of Mexico City and those from the Central Valley of Mexico are not different in terms of ethnic origin, i.e. mestizos.
3. Classical risk factors of atherothrombosis could be indicated under the form of Tables to avoid extensive description in the text including the way they were collected.
4. Lines 49-51, p.5, "one gene was involved in coagulation mechanism" vWA is a STR of intron 40 of the von Willebrand factor gene (Table S2) and not "the gene involved in coagulation mechanism" This tetranucleotide repeat is an excellent marker for genetic studies.
5. Page 10, lines 13-17. "Lipoprotein(a) [Lp(a)], an enhancer of foam cell production and a fibrinolytic inhibitor, is encoded by LPA, which genetic variants control both its isoforms size variation as plasma Lp(a) levels (10)" This sentence needs clarification. First, the main enhancer of foam cell production is LDL. Only when Lp(a) is considerably increased it could contribute to foam cell production via its LDL component. Second Lp(a) is a mix of LDL and apolipoprotein apo(a); LPA gene is the code for apo(a). Genetic variants of the LPA gene control isoforms size of apo(a) and thereby variation as plasma Lp(a) levels. Association of high Lp(a) levels and small isoform size is a well recognized factor of atherothrombosis. Was Lp(a) dosage and apo(a) isoform identification considered by authors? If not, it could be an important asset for further investigation in these populations.

Minor corrections

The sentence “The higher consumption of tobacco (> two pack/year) and alcohol (> 20 g/week) was presented within the CMA individuals.” needs correction.

Line 43: “smoking history, CVD’s family background, hypercholesterolemia, and PA » “PA” refers to LPA? or, I guess, Physical Activity..

Page 10

Lines 19-31 The whole paragraph should be revised and corrected.

The sentence “The atherogenic component of Lp(a) -apoprotein (a)-, is structurally homologous to plasminogen and competes for bind to its receptor as well as fibrin and fibrinogen,”

Please correct as Binding of apo(a) to fibrin does not promote free radicals generation (36-38).

Lp(a) stimulates the endothelial permeability and adhesion molecules production as well as proinflammatory cytokines release.

“besides to be an independent risk factor for cerebral and CV atherothrombosis (4, 39). Smaller and larger alleles than the LPA*8 have been associated with the prothrombotic activity, reinforcing the our findings of the contribution of LPA to atherothrombosis development (40

Review form: Reviewer 2

Is the manuscript scientifically sound in its present form?

Yes

Are the interpretations and conclusions justified by the results?

Yes

Is the language acceptable?

Yes

Is it clear how to access all supporting data?

Yes

Do you have any ethical concerns with this paper?

No

Have you any concerns about statistical analyses in this paper?

No

Recommendation?

Accept with minor revision (please list in comments)

Comments to the Author(s)

The authors of the study entitled: ALOX5, LPA , MMP9 and TPO genetic variants increase atherothrombosis susceptibility in middle-aged Mexican population, aimed to examine the role of genetic variants of ALOX5, IL6, LPA, MMP9, vWA, eNOS, HMOX1, NOX4, and TPO in atherothrombosis in Mexican population and found association of ALOX5, LPA, MMP9 and TPO variants. The study is interesting and worth of publishing.

Major comments

1. rs numbers of analyzed variants should given, names of all alleles, location within the gene, their functional effect should be given in introduction and association with diseases. Clearly give in tables allele and all genotypes. Methods applied should be described clearly, with references. Results, emphasize important revelations clearly, and give concrete not descriptive results.

2. The authors should avoid using confusing structure, and explain methods and protocols used clearly and simply. genotyping should be given more clearly, regardless the process in which they are involved in.

one term should be used in the whole manuscript polymorphism or variant

Association test should be written more clearly

3. Results should not be descriptive, than clearly emphasis the difference with statistical parameters

Table 1, family history, positive (%)

4. genome control- give explanation of the term and reference

5. Give results of HW disequilibrium in CME group, what is observed and what expected freq (p 0.0005) $\chi^2 = 0.002$ what is the format of p value?

6. multi loci comparison, how it was calculated by cumulative effect, gene-gene interaction, should be explained It is not clear

7. >0.05 should be changed to ns- non significant, keep consistency in presenting results through manuscript

8. instead loci, authors could use a term variants in the whole manuscript, they use different terms for the same thing and make confusion with that

9. Titles of the tables should be clear, concise and without sufficient information

10. Sentence Epidemiological studies,

cell cultures, and animal models have positively related the low numbers of tandem repeats on the promoter region of ALOX5 to an increased risk of CMA, reinforcing our findings (24-26, 30) is very confusing

Smaller and larger alleles than the LPA*8 have been associated with the pro thrombotic activity, reinforcing the our findings of the contribution of LPA to atherothrombosis development (40). It is not clear what is smaller or larger alleles, 11. instead of reinforcing authors should say which is in accordance with our results

12. MMP9 is mainly regulated at transcriptional level; polymorphisms located on the promoter region have been associated with MI, ischemic stroke, and coronary artery disease (38, 42), lung disorders and carcinoma should be added:

References, e.g.

Gene-environment interaction between the MMP9 C-1562T promoter variant and cigarette smoke in the pathogenesis of chronic obstructive pulmonary disease.

Stankovic M, Kojic S, Djordjevic V, Tomovic A, Nagorni-Obradovic L, Petrovic-Stanojevic N, Mitic-Milicic M, Radojkovic D.

Environ Mol Mutagen. 2016 Jul;57(6):447-54.

Matrix metalloproteinase-9 -1562T allele and its combination with MMP-2 -735 C allele are risk factors for breast cancer.

Rahimi Z, Yari K, Rahimi Z.

Asian Pac J Cancer Prev. 2015;16(3):1175-9.

13. collective gene participation, could be multifactorial nature of disease

14. results of genotypes should be given also for IL-6, vWA, eNOS, HMOX1, and NOX-4

15. reinforce, cornerston- too many repeats in the manuscript

16. Weaknesses were the modest sample size, sample size is quite satisfying and should not be mentioned as weakness

Authors could neglect limitations of the study mentioned.

17. title table 1. characteristics of the groups
 table 2 first columns give data of the groups than p-values of comparisons, title should be short and concise
 table 3 give OR (95% CI) format per column
18. Authors should Proof read the manuscript with native English editor

Minor comments

1. gene-variants, at-risk, multi-ethnic Forty-two, at-susceptibility, complex-diseases, etc. no need for dashes
2. Biological pathways, biochemical pathways
3. Page4 morbid-, population-, ancestral- ?
4. evaluated starting with the questions do you smoke or drink alcohol? evaluated by questionnaire
5. (units of alcohol consumed) * (standard drink unit) * (frequency of drink habits by week) this is confusing
6. The genetic structure for the nine polymorphism, this is confusing
7. Titles Genotyping assessment, should be Genotyping or Determination of genotypes
 Differences between the groups according to cardiovascular diseases could be Characteristics of (analyzed) groups
 Allele and genotype disparities among the studied groups, could be distribution of alleles and genotypes among groups
 Statistical population genetic parameters, very confusing (define group or population in paper)
 Multi-loci comparisons among the evaluated groups, could be Multi-loci comparisons
8. what is drastic decrease, avoid descriptive terms in the results
9. peoples, should be singular
 peopling ?
 [Lp(a)] ?

Decision letter (RSOS-190775.R0)

09-Jul-2019

Dear Dr GOMEZ,

The editors assigned to your paper ("ALOX5, LPA, MMP9 and TPO genetic variants increase atherothrombosis susceptibility in middle-aged Mexican population.") have now received comments from reviewers.

While both reviewers are relatively positive about publication of the paper, they both raise a substantive number of major comments regarding the presentation and analysis of the data. These will need careful consideration. We would like you to revise your paper in accordance with the referee suggestions which can be found below (not including confidential reports to the Editor). Please note this decision does not guarantee eventual acceptance.

Please submit a copy of your revised paper before 01-Aug-2019. Please note that the revision deadline will expire at 00.00am on this date. If we do not hear from you within this time then it will be assumed that the paper has been withdrawn. In exceptional circumstances, extensions may be possible if agreed with the Editorial Office in advance. We do not allow multiple rounds of revision so we urge you to make every effort to fully address all of the comments at this stage.

If deemed necessary by the Editors, your manuscript will be sent back to one or more of the original reviewers for assessment. If the original reviewers are not available, we may invite new reviewers.

- Data accessibility

If you wish to submit your supporting data or code to Dryad (<http://datadryad.org/>), or modify your current submission to dryad, please use the following link:
<http://datadryad.org/submit?journalID=RSOS&manu=RSOS-190775>

- Competing interests

- Authors' contributions

- Acknowledgements

- Funding statement

on behalf of Professor Joris Veltman (Associate Editor) and Steve Brown (Subject Editor)
openscience@royalsociety.org

Comments to Author:

Reviewers' Comments to Author:

Reviewer: 1

Comments to the Author(s)

The researchers looked for genetic associations with atherothrombosis development (clinical manifestations) in the protein-coding genomes of 408 unrelated Mexican individuals. 204 Individuals having clinical manifestations of atherothrombosis (n=204) were paired by age and gender to an equal number of blood donors without clinical manifestations. They then replicated the study in an independent group of 300 unrelated mestizo individuals, the so-called genome control, and in a group of 329 native subjects belonging to different ethnic groups. From the nine gene studied, the researchers discovered frequency differences in 4 of them: ALOX5, LPA, MMP9, and TPO. These were significantly more frequent in individuals with clinical atherothrombosis even after being adjusted by family history of cardiovascular disease, smoking, alcohol intake, and physical activity

Main concerns

1. In the abstract the authors included both patients and control groups in a collective of "1037 unrelated Mexican individuals". The real number of patients is 204, which had clinical atherothrombosis. There is an equal number of blood donors without clinical manifestations of atherothrombosis. There are also a genome control group of 300 unrelated individuals (equality in gender) and an ancestral control group of 329 from different ethnic origin. These differences

should be clearly indicated not only in the Abstract but also sufficiently detailed in section 3.1 Population studied.

2. I would like to understand which is the difference between the 204 asymptomatic blood donor controls and the so-called genome control group of 300 unrelated individuals (beside their geographical origin). I suppose that residents of Mexico City and those from the Central Valley of Mexico are not different in terms of ethnic origin, i.e. mestizos.

3. Classical risk factors of atherothrombosis could be indicated under the form of Tables to avoid extensive description in the text including the way they were collected.

4. Lines 49-51, p.5, "one gene was involved in coagulation mechanism"

vWA is a STR of intron 40 of the von Willebrand factor gene (Table S2) and not "the gene involved in coagulation mechanism" This tetranucleotide repeat is an excellent marker for genetic studies.

5. Page 10, lines 13-17. "Lipoprotein(a) [Lp(a)], an enhancer of foam cell production and a fibrinolytic inhibitor, is encoded by LPA, which genetic variants control both its isoforms size variation as plasma Lp(a) levels (10)" This sentence needs clarification. First, the main enhancer of foam cell production is LDL. Only when Lp(a) is considerably increased it could contribute to foam cell production via its LDL component. Second Lp(a) is a mix of LDL and apolipoprotein apo(a); LPA gene is the code for apo(a). Genetic variants of the LPA gene control isoforms size of apo(a) and thereby variation as plasma Lp(a) levels. Association of high Lp(a) levels and small isoform size is a well recognized factor of atherothrombosis. Was Lp(a) dosage and apo(a) isoform identification considered by authors? If not, it could be an important asset for further investigation in these populations.

Minor corrections

Page 8

The sentence "The higher consumption of tobacco (> two pack/year) and alcohol (> 20 g/week) was presented within the CMA individuals." needs correction.

Line 43: "smoking history, CVD's family background, hypercholesterolemia, and PA » "PA" refers to LPA? or, I guess, Physical Activity..

Page 10

Lines 19-31 The whole paragraph should be revised and corrected.

The sentence "The atherogenic component of Lp(a) -apoprotein (a)-, is structurally homologous to plasminogen and competes for bind to its receptor as well as fibrin and fibrinogen,"

Please correct as Binding of apo(a) to fibrin does not promote free radicals generation (36-38).

Lp(a) stimulates the endothelial permeability and adhesion molecules production as well as proinflammatory cytokines release.

"besides to be an independent risk factor for cerebral and CV atherothrombosis (4, 39). Smaller and larger alleles than the LPA*8 have been associated with the prothrombotic activity, reinforcing the our findings of the contribution of LPA to atherothrombosis development (40)

Reviewer: 2

Comments to the Author(s)

The authors of the study entitled: ALOX5, LPA , MMP9 and TPO genetic variants increase atherothrombosis susceptibility in middle-aged Mexican population, aimed to examine the role of

genetic variants of ALOX5, IL6, LPA, MMP9, vWA, eNOS, HMOX1, NOX4, and TPO in atherothrombosis in Mexican population and found association of ALOX5, LPA, MMP9 and TPO variants. The study is interesting and worth of publishing.

Major comments

1. rs numbers of analyzed variants should given, names of all alleles, location within the gene, their functional effect should be given in introduction and association with diseases. Clearly give in tables allele and all genotypes. Methods applied should be described clearly, with references. Results, emphasize important revelations clearly, and give concrete not descriptive results.

2. The authors should avoid using confusing structure, and explain methods and protocols used clearly and simply. genotyping should be given more clearly, regardless the process in which they are involved in.

one term should be used in the whole manuscript polymorphism or variant

Association test should be written more clearly

3. Results should not be descriptive, than clearly emphasis the difference with statistical parameters

Table 1, family history, positive (%)

4. genome control- give explanation of the term and reference

5. Give results of HW disequilibrium in CME group, what is observed and what expected freq (p 0.0005) $\chi^2 = 0.002$ what is the format of p value?

6. multi loci comparison, how it was calculated by cumulative effect, gene-gene interaction, should be explained It is not clear

7. >0.05 should be changed to ns- non significant, keep consistency in presenting results through manuscript

8. instead loci, authors could use a term variants in the whole manuscript, they use different terms for the same thing and make confusion with that

9. Titles of the tables should be clear, concise and without sufficient information

10. Sentence Epidemiological studies,

cell cultures, and animal models have positively related the low numbers of tandem repeats on the promoter region of ALOX5 to an increased risk of CMA, reinforcing our findings (24-26, 30) is very confusing

Smaller and larger alleles than the LPA*8 have been associated with the pro thrombotic activity, reinforcing the our findings of the contribution of LPA to atherothrombosis development (40). It is not clear what is smaller or larger alleles, 11. instead of reinforcing authors should say which is in accordance with our results

12. MMP9 is mainly regulated at transcriptional level; polymorphisms located on the promoter region have been associated with MI, ischemic stroke, and coronary artery disease (38, 42), lung disorders and carcinoma should be added:

References, e.g.

Gene-environment interaction between the MMP9 C-1562T promoter variant and cigarette smoke in the pathogenesis of chronic obstructive pulmonary disease.

Stankovic M, Kojic S, Djordjevic V, Tomovic A, Nagorni-Obradovic L, Petrovic-Stanojevic N, Mitic-Milicic M, Radojkovic D.

Environ Mol Mutagen. 2016 Jul;57(6):447-54.

Matrix metalloproteinase-9 -1562T allele and its combination with MMP-2 -735 C allele are risk factors for breast cancer.

Rahimi Z, Yari K, Rahimi Z.

Asian Pac J Cancer Prev. 2015;16(3):1175-9.

13. collective gene participation, could be multifactorial nature of disease

14. results of genotypes should be given also for IL-6, vWA, eNOS, HMOX1, and NOX-4
15. reinforce, cornerston- too many repeats in the manuscript
16. Weaknesses were the modest sample size, sample size is quite satisfying and should not be mentioned as weekness
Authors could negelect limitations of the study mentioned.
17. title table 1. characteristics of the groups
table 2 first columns give data of the groups than p-values of comparisons, title should be short and concise
table 3 give OR (95% CI) format per column
18. Authors should Proof read the manuscript with native English editor

Minor comments

1. gene-variants, at-risk, multi-ethnic Forty-two, at-susceptibility, complex-diseases, etc. no need for dashes
2. Biological pathways, biochemical pathways
3. Page4 morbid-, population-, ancestral- ?
4. evaluated starting with the questions do you smoke or drink alcohol? evaluated by questionnaire
5. (units of alcohol consumed) * (standard drink unit) * (frequency of drink habits by week) this is confusing
- 6.The genetic structure for the nine polymorphism, this is confusing
7. Titles Genotyping assessment, should be Genotyping or Determination of genotypes
Differences between the groups according to cardiovascular diseases could be Characteristics of (analyzed) groups
Allele and genotype disparities among the studied groups, could be distribution of alleles and genotypes among groups
Statistical population genetic parameters, very confusing (define group or population in paper)
Multi-loci comparisons among the evaluated groups, could be Multi-loci comparisons
8. what is drastic decrease, avoid descriptive terms in the results
9. peoples, should be singular
peopling ?
[Lp(a)] ?

Editorial Office Comments to Author:

For information about language editing services endorsed by the Royal Society, please follow the link below:

<https://royalsociety.org/journals/authors/language-polishing/>

Author's Response to Decision Letter for (RSOS-190775.R0)

See Appendix A.

RSOS-190775.R1 (Revision)

Review form: Reviewer 1

Is the manuscript scientifically sound in its present form?

Yes

Are the interpretations and conclusions justified by the results?

Yes

Is the language acceptable?

Yes

Do you have any ethical concerns with this paper?

No

Have you any concerns about statistical analyses in this paper?

No

Recommendation?

Accept with minor revision (please list in comments)

Comments to the Author(s)

The authros indicate in section 3.1 first paragraph "Classical risk factors such as diabetes type 2, hypercholesterolemia, body mass index, age, and family history of CVDs were obtained from the clinical records of these 408 individuals"

My comment: healthy controls were selected among healthy blood donors and they should not have a clinical record at the moment of sampling. Please, modify as necessary.

I have no further coments or questions

Review form: Reviewer 2

Is the manuscript scientifically sound in its present form?

Yes

Are the interpretations and conclusions justified by the results?

Yes

Is the language acceptable?

Yes

Do you have any ethical concerns with this paper?

No

Have you any concerns about statistical analyses in this paper?

No

Recommendation?

Accept with minor revision (please list in comments)

Comments to the Author(s)

The authors corresponded well to the reviewers' suggestions. Just some more minor comments:

1. page 10/4 MMP9 italic
2. page 12/39-13/11 No need for mentioning questions from questioner, just say that in smokers/ex-smokers cumulative cigarette consumption is expressed by pack-years (standard well-known unit) till the occurrence of disease. The same is for alcohol consumption.
3. Table 2, as previously suggested first give columns with subjects than p values for all comparisons, footnote is not corresponding to the symbols in the table, give the no. and % of reference category, for LPA, usually denoted as 1, or eliminate non-carriers, genotypes are usually presented as X/X, or X/(X)...what is X in ALOX5?
4. indicate genes shown in tables s7-15
5. s16-23 column 1 is only allele, genes are indicated in subsequent columns
6. hoja worksheet is empty
7. Table 3. Clinical manifestation of AT is only above yes and no
Footnotes are not corresponding to the symbols in the table,
OR and 95% CI can be given in one column using OR (95% CI) format,
keep one or two decimals in the tables consistently
give the no of subjects and %-age in yes/no column, and for LPA eliminate non-carriers as they are not needed.
8. Suggestions:
Title Table 1 Characteristics of study participants
Title Table 2. and 3 Association of genetic risks with AT. Genetic risks is not good term as it predisposes the risk of the diseases, as already established, it could be changed to genetic factors.
In table 2 only p values are shown while in table 3 only ORs, both values should be shown in both tables. Keep consistency and simplicity within the whole manuscript.

Decision letter (RSOS-190775.R1)

12-Nov-2019

Dear Dr Gomez,

On behalf of the Editors, I am pleased to inform you that your Manuscript RSOS-190775.R1 entitled "ALOX5, LPA, MMP9 and TPO genetic variants increase atherosclerosis susceptibility in middle-aged Mexicans." has been accepted for publication in Royal Society Open Science subject to minor revision in accordance with the referee suggestions. Please find the referees' comments at the end of this email.

The reviewers and Subject Editor have recommended publication, but also suggest some minor revisions to your manuscript. Therefore, I invite you to respond to the comments and revise your manuscript.

- Ethics statement

- Data accessibility

If you wish to submit your supporting data or code to Dryad (<http://datadryad.org/>), or modify your current submission to dryad, please use the following link:
<http://datadryad.org/submit?journalID=RSOS&manu=RSOS-190775.R1>

- Competing interests

- Authors' contributions

- Acknowledgements

- Funding statement

Because the schedule for publication is very tight, it is a condition of publication that you submit the revised version of your manuscript before 21-Nov-2019. Please note that the revision deadline will expire at 00.00am on this date. If you do not think you will be able to meet this date please let me know immediately.

Kind regards,
Lianne Parkhouse
Editorial Coordinator
Royal Society Open Science
openscience@royalsociety.org

on behalf of the Associate Editor, and Professor Steve Brown (Subject Editor)
openscience@royalsociety.org

Associate Editor Comments to Author:

Your manuscript has returned from peer review, and the referees' comments can be found below. Specifically, please address the comment made by Referee #1 on the sampling, and the minor comments made by Referee #2 when submitting your revision.

Reviewer comments to Author:

Reviewer: 1

Comments to the Author(s)

The authors indicate in section 3.1 first paragraph "Classical risk factors such as diabetes type 2, hypercholesterolemia, body mass index, age, and family history of CVDs were obtained from the clinical records of these 408 individuals"

My comment: healthy controls were selected among healthy blood donors and they should not have a clinical record at the moment of sampling. Please, modify as necessary.

I have no further comments or questions

Reviewer: 2

Comments to the Author(s)

The authors corresponded well to the reviewers' suggestions. Just some more minor comments:

1. page 10/4 MMP9 italic
 2. page 12/39-13/11 No need for mentioning questions from questioner, just say that in smokers/ex-smokers cumulative cigarette consumption is expressed by pack-years (standard well-known unit) till the occurrence of disease. The same is for alcohol consumption.
 3. Table 2, as previously suggested first give columns with subjects than p values for all comparisons, footnote is not corresponding to the symbols in the table, give the no. and % of reference category, for LPA, usually denoted as 1, or eliminate non-carriers, genotypes are usually presented as X/X, or X/(X)...what is X in ALOX5?
 4. indicate genes shown in tables s7-15
 5. s16-23 column 1 is only allele, genes are indicated in subsequent columns
 6. hoja worksheet is empty
 7. Table 3. Clinical manifestation of AT is only above yes and no
- Footnotes are not corresponding to the symbols in the table,
 OR and 95% CI can be given in one column using OR (95% CI) format,
 keep one or two decimals in the tables consistently
 give the no of subjects and %-age in yes/no column, and for LPA eliminate non-carriers as they are not needed.
8. Suggestions:
- Title Table 1 Characteristics of study participants
 Title Table 2. and 3 Association of genetic risks with AT. Genetic risks is not good term as it predisposes the risk of the diseases, as already established, it could be changed to genetic factors.
 In table 2 only p values are shown while in table 3 only ORs, both values should be shown in both tables. Keep consistency and simplicity within the whole manuscript.

Author's Response to Decision Letter for (RSOS-190775.R1)

See Appendix B.

Decision letter (RSOS-190775.R2)

28-Nov-2019

Dear Dr GOMEZ,

It is a pleasure to accept your manuscript entitled "ALOX5, LPA, MMP9 and TPO genetic variants increase atherothrombosis susceptibility in middle-aged Mexicans." in its current form for publication in Royal Society Open Science. The comments of the reviewer(s) who reviewed your manuscript are included at the foot of this letter.

on behalf of Prof Steve Brown (Subject Editor)
openscience@royalsociety.org

Appendix A

CENTRO DE INVESTIGACIÓN Y DE ESTUDIOS AVANZADOS DEL INSTITUTO POLITÉCNICO NACIONAL

August 5, 2019.

Professor Jeremy Sanders
Editor-in-Chief
Royal Society Open Science

Dear Sir,

Attached is the manuscript by Camacho-Mejorado *et al.* entitled "**ALOX5, LPA, MMP9 and TPO genetic variants increase atherothrombosis susceptibility in middle-aged Mexicans**", which is being submitted for possible publication as *Research*.

Firstly, we want to acknowledge the suitable criticisms of the reviewers, which have been incorporated in this new version. Major changes include:

- A detailed explained about the differences between the healthy group and the genomic control was added in the **Material and Methods** section.
- The incorporation of a new table (Table S1) in the **Supplementary material** section summarising the clinical risk factors.
- The incorporation of a new table (Table S3) in the **Supplementary material** section with general information about the studied polymorphisms.
- A modification in the **title**. We have eliminated the word population.

Further, we have considered and replicated all the reviewers' suggestions, which have been supported with a brief discussion accompanied by references.

We hope our manuscript be of interest to be published in your prestigious Journal.

I am looking forward to hearing from you soon.

Rocío Gómez, MSc, PhD
Department of Toxicology, Cinvestav-IPN
e-mail: mrgomez@cinvestav.mx
Office phone: +52 5747-3800 ext.6770
Fax: +52 5747-3395

**CENTRO DE INVESTIGACIÓN Y DE ESTUDIOS AVANZADOS
DEL INSTITUTO POLITÉCNICO NACIONAL**

Responses to the reviewers' comments.

Reviewer 1.

We appreciate all your comments.

Main concerns.

Comment 1.

“In the abstract the authors included both patients and control groups in a collective of “1037 unrelated Mexican individuals”. The real number of patients is 204, which had clinical atherothrombosis. There is an equal number of blood donors without clinical manifestations of atherothrombosis. There are also a genome control group of 300 unrelated individuals (equality in gender) and an ancestral control group of 329 from different ethnic origin. These differences should be clearly indicated not only in the Abstract but also sufficiently detailed in section.”

Comment 2.

“I would like to understand which is the difference between the 204 asymptomatic blood donor controls and the so-called genome control group of 300 unrelated individuals (beside their geographical origin). I suppose that residents of Mexico City and those from the Central Valley of Mexico are not different in terms of ethnic origin, i.e. mestizos.”

*We agree with your criticism. Hence, we have replicated your suggestions both in **Abstract** as in the **Material and Methods** section. Notably, in the subheading **3.1 Characteristics of the studied groups**, we have explained, in detail, the differences between the healthy group and the genomic control (GC). Besides, we have included information about the characteristics of the GC pointed out its utility in populations genetically heterogeneous such as the Mexican one. The references that support the used information appear in the main text.*

Comment 3.

“Classical risk factors of atherothrombosis could be indicated under the form of Tables to avoid extensive description in the text including the way they were collected.”

*Thank you so much for this suggestion. Following your recommendation, a new table (**Table S1**) in the **Supplementary material section** has been added. In this table, the indicators and parameters of the two clinical risk factors (diabetes type 2 and hypercholesterolemia) have been summarised. Regarding the other classical risk factors (dietary, physical activity, smoking habit and alcohol intake) we have summed up their information to be brief given that we have not considered indicating all risk factors in the form of a table.*

**CENTRO DE INVESTIGACIÓN Y DE ESTUDIOS AVANZADOS
DEL INSTITUTO POLITÉCNICO NACIONAL**

Comment 4.

Lines 49-51, p.5, “one gene was involved in coagulation mechanism” vWA is a STR of intron 40 of the von Willebrand factor gene (Table S2) and not “the gene involved in coagulation mechanism” This tetranucleotide repeat is an excellent marker for genetic studies.

Comment 5.

Page 10, lines 13-17. “Lipoprotein(a) [Lp(a)], an enhancer of foam cell production and a fibrinolytic inhibitor, is encoded by LPA, which genetic variants control both its isoforms size variation as plasma Lp(a) levels (10)” This sentence needs clarification. First, the main enhancer of foam cell production is LDL. Only when Lp(a) is considerably increased it could contribute to foam cell production via its LDL component. Second Lp(a) is a mix of LDL and apolipoprotein apo(a); LPA gene is the code for apo(a). Genetic variants of the LPA gene control isoforms size of apo(a) and thereby variation as plasma Lp(a) levels. Association of high Lp(a) levels and small isoform size is a well recognized factor of atherothrombosis. Was Lp(a) dosage and apo(a) isoform identification considered by authors? If not, it could be an important asset for further investigation in these populations.

We agree with these two comments. These inaccuracies have been clarified.

Minor corrections.

Thank you for your observations; all of them have been clarified in the main text.

**CENTRO DE INVESTIGACIÓN Y DE ESTUDIOS AVANZADOS
DEL INSTITUTO POLITÉCNICO NACIONAL**

Reviewer 2.

We appreciate all your comments.

Major comments.

Comment 1.

“rs numbers of analyzed variants should given, names of all alleles, location within the gene, their functional effect should be given in introduction and association with diseases. Clearly give in tables allele and all genotypes. Methods applied should be described clearly, with references. Results, emphasize important revelations clearly, and give concrete not descriptive results.”

Thank you so much for this suggestion.

In this setting, we have added a new table (Table S3) in the **Supplementary material** section. Of note, reference SNP (rs), is an identification tag assigned by NCBI to a group of SNPs, although this ID has been used in all types of variants (<https://www.ncbi.nlm.nih.gov/books/NBK44417/>).

Out of nine short tandem repeat explored, three (ALOX5, MMP9 and TPO) have been identified with rs number. About ALOX5 gene polymorphism, the rs59439148 refers an insertion/deletion of some alleles (<https://www.ncbi.nlm.nih.gov/snp/rs59439148>). Nonetheless, different authors (cited below) have assigned the rs59439148 to the tandem repeat polymorphism in the promoter region of ALOX5; we have followed this criterion.

References:

Gammelmark A, et al. 2017. *J Nutr* **147**, 1340-1347, doi: [10.3945/jn.117.247569](https://doi.org/10.3945/jn.117.247569)

Gammelmark A, et al. 2016. *PLoS ONE* **11**, e0167217, doi: [10.1371/journal.pone.0167217](https://doi.org/10.1371/journal.pone.0167217)

Mougey E et al. 2013. *Clin Exp Allergy* **43**, 512-520, doi: [10.1111/cea.12076](https://doi.org/10.1111/cea.12076)

Lang JE, et al. 2013. *Contemp Clin Trials* **34**, 326-335, doi: [10.1016/j.cct.2012.12.009](https://doi.org/10.1016/j.cct.2012.12.009)

Similarly, MMP9 polymorphism has been referred using the rs2234681 (references below). (<https://www.ncbi.nlm.nih.gov/snp/?term=rs2234681>)

References:

Fiotti N, et al. 2018. *J Vac Surg*. **67**, 1727-1735, doi: [10.1016/j.jvs.2017.09.047](https://doi.org/10.1016/j.jvs.2017.09.047)

Marson BP, et al. 2012. *Clin Chim Acta*. **414**, 46-51, doi: [10.1016/j.cca.2012.08.014](https://doi.org/10.1016/j.cca.2012.08.014)

**CENTRO DE INVESTIGACIÓN Y DE ESTUDIOS AVANZADOS
DEL INSTITUTO POLITÉCNICO NACIONAL**

Luizon MR, et al. 2012. Hypertens Res. 35, 917-921, doi: [10.1038/hr.2012.60](https://doi.org/10.1038/hr.2012.60)

Metzger IF, et al. 2012. DNA Cell Biol. 31, 504-510, doi: [10.1089/dna.2011.1388](https://doi.org/10.1089/dna.2011.1388)

Belo VA, et al. 2012. Int J Obs (Lond). 36, 69-75, doi: [10.1038/ijo.2011.169](https://doi.org/10.1038/ijo.2011.169)

Laccini R, et al. 2010. Clin Chim Acta. 411, 1940-1944, doi: [10.1016/j.cca.2010.08.008](https://doi.org/10.1016/j.cca.2010.08.008)

TPO tandem repeat is identified with the rs13422969. Nonetheless, it refers to the (ATTG)₉ allele. Thus, this ID has not used in the table presented herein. We have completed the information about the polymorphisms explored, including other data such as UniSTS, GeneBank accession and Ensembl reference. This information was included briefly, in the introduction.

Comment 2.

“The authors should avoid using confusing structure, and explain methods and protocols used clearly and simply. genotyping should be given more clearly, regardless the process in which they are involved in one term should be used in the whole manuscript polymorphism or variant Association test should be written more clearly .”

Thank you so much for this suggestion. We have clarified the methods and protocols used to genotyped the studied polymorphisms. Further, the whole manuscript was homogenised with the term polymorphisms.

Comment 3.

“Results should not be descriptive, than clearly emphasis the difference with statistical parameters Table 1, family history, positive (%)”.

Thank you so much for this criticism. We have incorporated your suggestions.

Comment 4.

“genome control- give explanation of the term and reference”

We agree with these comments. Thus, this has been clarified.

Comment 5.

“Give results of HW disequilibrium in CME group, what is observed and what expected freq (p 0.0005)± = 0.002 what is the format of p value?”

*Thank you for this comment. The p-value format has been unified and the observed and expected heterozygous values were incorporated in the **Results** section (**4.4 Genetic statistical parameters**).*

**CENTRO DE INVESTIGACIÓN Y DE ESTUDIOS AVANZADOS
DEL INSTITUTO POLITÉCNICO NACIONAL**

Comment 6.

multi loci comparison, how it was calculated by cumulative effect, gene-gene interaction, should be explained It is not clear.

*Thank you so much for this suggestion. It was clarified in the **Material and Methods** section (3.5.2 Genetic statistics) where we have substituted multi loci comparisons by multiallelic combination.*

Comments 7, 8 and 9.

7. >0.05 should be changed to ns- non significant, keep consistency in presenting results through manuscript

8. instead loci, authors could use a term variants in the whole manuscript, they use different terms for the same thing and make confusion with that

9. Titles of the tables should be clear, concise and without sufficient information

Thank you for these suggestions; all of them have been clarified in the main text.

Comments 10 and 11.

10. Sentence Epidemiological studies, cell cultures, and animal models have positively related the low numbers of tandem repeats on the promoter region of ALOX5 to an increased risk of CMA, reinforcing our findings (24-26, 30) is very confusing Smaller and larger alleles than the LPA*8 have been associated with the pro thrombotic activity, reinforcing the our findings of the contribution of LPA to atherothrombosis development (40). It is not clear what is smaller or larger alleles, 11. instead of reinforcing authors should say which is in accordance with our results.

Thank you for these suggestions; all of them have been clarified in the main text.

Comment 12.

MMP9 is mainly regulated at transcriptional level; polymorphisms located on the promoter region have been associated with MI, ischemic stroke, and coronary artery disease (38, 42), lung disorders and carcinoma should be added...

We appreciate this suggestion; we have incorporated the references suggested.

**CENTRO DE INVESTIGACIÓN Y DE ESTUDIOS AVANZADOS
DEL INSTITUTO POLITÉCNICO NACIONAL**

Comment 13.

collective gene participation, could be multifactorial nature of disease

Thank you so much, we have done reference to this multifactorial nature of the disease in the main text.

Comment 14.

results of genotypes should be given also for IL-6, vWA, eNOS, HMOX1, and NOX-4

*We agree with your criticism; we have incorporated your suggestion in the **Results** section, in the last paragraph of **4.2 Distribution of allele and genotypes in CMA and healthy groups**.*

Comments 15 and 17.

15. reinforce, cornerston- too many repeats in the manuscript

17. title table 1. characteristics of the groups table 2 first columns give data of the groups than p-values of comparisons, title should be short and concisetable 3 give OR (95% CI) format per column

We agree with your criticisms and we have added all your suggestions.

Regarding table 2, we have modified the title. Your comment about the first column, we considered that give the genes studied, the risk categories and then the data obtained in the groups with and without clinical manifestations of atherothrombosis depicted, clearly, the data of the groups.

Comments 16.

Weaknesses were the modest sample size, sample size is quite satisfying and should not be mentioned as weakness

Authors could neglect limitations of the study mentioned.

Thank you so much for this comment. The paragraph was removed.

Minor comments.

Thank you for your observations; all of them have been clarified in the main text.

Appendix B

CENTRO DE INVESTIGACIÓN Y DE ESTUDIOS AVANZADOS DEL INSTITUTO POLITÉCNICO NACIONAL

November 15, 2019.

Professor Jeremy Sanders
Editor-in-Chief
Royal Society Open Science

Dear Sir,

Attached is the final version of our manuscript (RSOS-190775.R2) entitled "*ALOX5, LPA, MMP9 and TPO genetic variants increase atherothrombosis susceptibility in middle-aged Mexicans*", which has been accepted for publication as *Research* in your prestigious journal.

We restate our acknowledgements to the Reviewers' comments, which have increased the quality of our manuscript. Overall, all the referees' suggestions have been addressed (next sheet). Likewise, ethics statement, data accessibility, competing interests, Authors' contributions, acknowledgements and funding statement have been attached following the format suggested.

Kind regards,

Rocío Gómez, MSc, PhD
Department of Toxicology, Cinvestav-IPN
e-mail: mrgomez@cinvestav.mx
Office phone: +52 5747-3800 ext.6770
Fax: +52 5747-3395

**CENTRO DE INVESTIGACIÓN Y DE ESTUDIOS AVANZADOS
DEL INSTITUTO POLITÉCNICO NACIONAL**

Responses to the reviewers' comments.

Referee 1.

Thank you for your observations; all of them have been attended in the main text.

Referee 2.

We appreciate all your comments.

Minor comments.

Comment 1.

“page 10/4 MMP9 italic”

Thank you for this observation; it has been modified in the Summary section.

Comment 2.

“page 12/39-13/11 No need for mentioning questions from questioner, just say that in smokers/ex-smokers cumulative cigarette consumption is expressed by pack-years (standard well-known unit) till the occurrence of disease. The same is for alcohol consumption”.

Thank you so much for this proposal. We have attended all your suggestions, which are exhibited in section 3.1.

Comment 3.

“Table 2, as previously suggested first give columns with subjects than p values for all comparisons, footnote is not corresponding to the symbols in the table, give the no. and % of reference category, for LPA, usually denoted as 1, or eliminate non-carriers, genotypes are usually presented as X/X, or X/(X)...what is X in ALOX5?”

Thank you for your criticism. Nonetheless, the modification of the order of the columns could cause some confusion. Thus, we have modified the table to be more comprehensible, representing the differences between atherothrombosis cases versus healthy controls. Further, the reference category for each locus was included. Regarding the symbols (including the “X” in ALOX5) these have been modified and clarified. About your suggestion to eliminate the non-carriers, we have changed the terminology of the allelic representation. Nonetheless, the non-carriers condition has not been eliminated, given that this condition is the risk one.

Comment 4.

“indicate genes shown in tables s7-15”

Thank you so much for these suggestions; all of them have been clarified.

**CENTRO DE INVESTIGACIÓN Y DE ESTUDIOS AVANZADOS
DEL INSTITUTO POLITÉCNICO NACIONAL**

Comment 5.

“s16-23 column 1 is only allele, genes are indicated in subsequent columns”

You are totally right. We have modified this mistake.

Comment 6.

“hoja worksheet is empty”

Thank you for this criticism; this worksheet has been deleted.

Comment 7.

“Table 3. Clinical manifestation of AT is only above yes and no Footnotes are not corresponding to the symbols in the table, OR and 95% CI can be given in one column using OR (95% CI) format, keep one or two decimals in the tables consistently give the no of subjects and %-age in yes/no column, and for LPA eliminate non-carriers as they are not needed”.

We appreciate all your comments.

Akin to table 2, table 3 has been modified depicting in this case, only the differences between atherothrombosis cases versus genomic control. Also, we have checked the footnotes to agree with the table. The OR and 95% CI are presenting in one column format now, maintaining two decimals in all tables consistently. About the suggestion to include % age in the yes/no column, we thought this is not necessary given that the atherothrombosis cases and the healthy controls have been matched by age and sex. The suggestion about to eliminate the LPA non-carriers has been clarified in comment 3. Given the modifications made to Table 2 and Table 3, the main text was reviewed and adapted to these them.

Comment 8.

Title Table 1 Characteristics of study participants.

This suggestion has been considered. Thank you so much.

Title Table 2 and 3 Association of genetic risks with AT. Genetic risks is not good term as it predisposes the risk of the diseases, as already established, it could be changed to genetic factors.

You are right. Thus, we have modified the term “genetic risk” for “genetic factors”. Thank you so much for this suggestion.

In table 2 only p values are shown while in table 3 only ORs, both values should be shown in both tables. Keep consistency and simplicity within the whole manuscript.

**CENTRO DE INVESTIGACIÓN Y DE ESTUDIOS AVANZADOS
DEL INSTITUTO POLITÉCNICO NACIONAL**

Thank you so much for your criticism. We have modified these aspects keeping the consistency in all the tables.